# Introduction of artificial plants has no detrimental or beneficial effects on laboratory zebrafish husbandry but limits available swimming space

**Aymene Youcef Krachni, Richard Busch, Indigo Brakus, Angelina Schumann, Jenny Wilzopolski, Nils Ohnesorge** *

German Federal Institute for Risk Assessment (BfR), Department German Centre for the Protection of Laboratory Animals and Experimental Toxicology, Berlin, Germany

* nils.ohnesorge@bfr.bund.de

## Abstract

There is a broad consensus that husbandry conditions of laboratory animals need constant improvement to guarantee optimal animal welfare and research data quality. Zebrafish (*Danio rerio*) as one of the main animal models in biomedicine and toxicology are currently kept in barren tanks in most experimental setups, as well as in animal husbandry. Structural enrichment with artificial plants is discussed at the moment as a potential refinement measure to provide a more diverse environment. Other reports have shown that this can reduce stress or anxiety and improve cognitive abilities, survival rate and fertility in these animals. Still, concerns remain regarding its long-term benefits and drawbacks. Therefore, we introduced artificial plants in our husbandry tanks, and evaluated over a one-year period if in our specific system the benefits would outweigh the risks. When we compared pairwise 16 tanks that were either non-enriched or enriched with artificial plants, we saw no significant difference in terms of zebrafish survival rate during rearing, sex ratio, fertility or pathogen burden. When analyzing zebrafish behavior in their 8 L home-tanks, we saw statistically significant avoidance of the area close to the plants and a place preference for the open water in the middle or opposite side of the tank. This effect got more pronounced at lower holding densities. In summary, we found that introducing structural enrichment to our specific zebrafish facility carried low cost and no detrimental effects for the animals but a reduction of their free-swimming space. At the same time benefits were difficult to determine in our readouts as the survival rates of our fish were already very high without structural enrichment. We would like to encourage others to prepare similar forms of facility reports regarding enrichment to ensure a broader discussion on their potential long-term benefits in zebrafish husbandry systems.

**Data availability statement:** All relevant data are within the paper and its Supporting Information files or will be available from the BioImage Archive under accession number S-BIAD2980.

**Funding:** The author(s) received no specific funding for this work.

**Competing interests:** The authors have declared that no competing interests exist.

## Introduction

Zebrafish (*Danio rerio*) is among the most successful animal models worldwide. Publication numbers increased by 75% in the past ten years (according to PubMed with 2814 publications in 2014 and 4949 in 2024, search term "zebrafish", accessed on 29th September, 2025). Despite its popularity in many research areas like biomedicine and toxicology, little is known about the optimal husbandry conditions for laboratory zebrafish. In the past, mainly barren tanks were used to improve hygiene standards and efficacy. Nowadays, the importance of zebrafish husbandry conditions and especially the need for improvement is increasingly recognized [1,2]. The reasons are on the one hand ethical obligations for best possible animal welfare as formulated in the EU directive 2010/63, and secondly, the acknowledgement of the role of environmental factors on data quality and reproducibility [3]. Hence, researchers should aim for a better understanding of zebrafish needs. As a first step, European legislation was recently updated to include a restricted range of permitted water parameters, acknowledging this process [4].

A way for improving husbandry conditions could be by providing appropriate environmental enrichment with the aim to increase the animal's environmental complexity via structural, dietary, social or cognitive measures [5]. Structural enrichment is defined as modification of the animal's habitat or the addition of physical structures or objects to promote natural behavior and increase overall well-being. Among others, reduced anxiety and stress as indicated by lower cortisol levels, better memory, learning and survivorship were reported [6–8]. In contrast, effects on aggression differed among studies, where both an increase and a decrease were observed [6]. Despite these encouraging findings, structural enrichment is still lacking similar acceptance as other forms of enrichment like visual or social enrichment or use of live feed that are increasingly common [9]. An international survey among zebrafish facilities that looked into this aspect reported widespread concerns regarding high workload, additional costs, consistency of scientific results and increased risk of pathogen growth. As a consequence, only the minority of zebrafish facilities have implemented structural enrichment or planned to do so [9]. Unfortunately, more recent data is lacking, but reported implementation in method sections of zebrafish publications remains low with a single mention of any form of refinement or enrichment in the 100 most recent research articles listed on PubMed employing zebrafish (search term "zebrafish", accessed on 8th July, 2025, see supplement S1 Table for details). Apparently, for many zebrafish facilities the feared risks or costs still outweigh the potential benefits and additionally the lack of reporting impedes change. One of the main problems seems to be that the potential benefits have been less tangible than improved water quality or better standardization of water parameters and that uncertainties remain regarding transferability of individual studies to standard husbandry settings.

Results from studies were single or pair-housed fish were used instead of shoaling groups and that were transferred from their home-tank to a novel environment [10,11] or where short-term effects of enrichment were observed in few days or weeks, might not be representative for a husbandry setting. In the systematic review

of Gallas-Lopez et al. on effects of structural enrichment only 3 out of 27 studies included were performed over a longer time frame than 3 months, while 21 were conducted for less than a month [12]. Therefore, further investigation is needed whether structural enrichment maintains its positive effects over time or whether constant changes are necessary to avoid habituation. Nevertheless, the systematic review came to the conclusion that, despite many confounding factors, the overall positive effects of structural enrichment remain [12].

Different forms of structural enrichment have been tested with varying degrees of success. While promising results were achieved with gravel substrate, artificial plants or similar simulations of vegetation, more artificial objects like an arrangement of glass rods or airstones were not beneficial or even counterproductive [13,14]. In addition, these structures inside the tank should not restrict free swimming behavior, block the view on the fish for daily health inspection, reduce water quality or induce territorial aggression between fish [15]. Therefore, the structural enrichment has to be provided in a way that the benefits outweigh the risks [16]. Observations from the natural habitat of the fish species used should guide the decisions on how to structure the tank environment [1].

Overall, the challenge remains to find suitable readouts to determine the welfare state of the fish, beyond the absence of illness. Here, among others, methods to investigate choice preference, levels of stress or anxiety, improved learning and memory, exploration behavior, aggression or fecundity have been employed [6,16]. As additional requirements for standard husbandry settings these methods should be non-invasive and performed in their home-tanks to avoid any additional stress or burden for the fish. In this way, they can be carried out regularly as part of the welfare assessment. Survivorship and sex ratio, for example, are such parameters that are constantly recorded in animal facilities and are easily available to assess enrichment effects [8].

In summary, zebrafish are a shoaling species that prefers open-water areas and evolved with various forms of vegetation in their natural habitat. They are highly adaptable and can live under a wide range of conditions which helped to promote their status as one of the major research models. It is necessary to optimize their laboratory husbandry conditions with structural enrichment as a form of refinement that could improve animal welfare and research data quality. But widespread implementation is still lacking as published studies on its effects are still lacking the robustness needed for possible long-term benefits to outweigh concerns for transferability to standard husbandry, pathogen risks and cost. To address these concerns and close the knowledge gaps, we documented the introduction of artificial plants in our facility and compared its effects with our standard husbandry conditions in all forms of readouts that we had already established as relevant for our research. We documented the results of fish survival and sex ratio after rearing, mating success, pathogen burden in the water and behavior of fish in their home-tanks with or without structural enrichment over the period of one year. Therefore, our aim for this study was to investigate whether positive or negative effects of structural enrichment could be detected in our facility-specific setting under long-term husbandry conditions in home-tanks, without any additional burden for the fish and low additional cost or work-load for the facility.

## Methods

The facility report was prepared and conducted according to PREPARE, ARRIVE and FAIR guidelines [17–19]. Raw data are available in the BioStudies database (http://www.ebi.ac.uk/biostudies) under accession number S-BIAD2980 [20].

## Animals

The following fish lines were used: *elavl3:H2B-GCaMP6s^jf5* (tank pairs 1 and 2), Tübingen wildtype (WT TU, tank pairs 3 and 4), *HuC:GCaMP5g^a4598* (tank pair 5), *HuC:Gal4/UAS:RFP* (tank pair 6), *mitfa^w2* (tank pair 7) and Tüpfel-Longfin wildtype (WT TL, tank pair 8). All lines except WT TL were maintained with Tübingen wildtype genetic background. In addition, *elavl3:H2B-GCaMP6s*, *HuC:GcaMP5g* and *HuC:Gal4/UAS:RFP* were in a *mitfa^-/-* background. No more than 3 generations of sibling matings were performed before lines were outcrossed against TU wildtype or *mitfa^-/-* lines with high genetic

heterogeneity to avoid inbreeding depression. TU and TL wildtype stocks are re-imported from the European Zebrafish Resource Center (EZRC) every 3 years to avoid creation of sublines.

## Environmental parameters and health status

Adult zebrafish were kept in 3.5 L or 8 L tanks with gravel pictures underneath, not segregated by sex at a holding density of about 5 fish/L in a Standalone husbandry rack (Tecniplast, Active Blue) with copper-free, recirculating water and a daily water exchange of 10–20%. Reverse osmosis was used for tap water purification from copper free piping, followed by reconstitution with sea salt. Water parameters were maintained at 28 °C, pH 7.5, conductivity 800 µS/cm, oxygen levels >7 mg/L, carbon dioxide levels 1–6 mg/L, carbonate hardness 2–5 °dH, total hardness 2–6 °dH, ammonia <0.05 mg/L, nitrite <0.02 mg/L and nitrate <25 mg/L [21]. All water parameters were either monitored continuously (temperature, pH, conductivity), daily (oxygen, carbon dioxide) or at least once a week (carbonate hardness, total hardness, ammonia, nitrite, nitrate). Room temperature was 24–25 °C, light intensity 500 lux in front of the tanks and 10–20 lux at the back of the tanks. Illumination was set to 10 hours of darkness followed by 30 min of dusk with increasing intensity to the maximum of 500 lux for 13 hours, followed by 30 minutes of decreasing light to darkness again. The adult fish were fed twice daily dry food (Sparos 400–600) *ad libitum* and once per day artemia (Sanders). Visual health checks were performed daily during feeding based on our approved care sheet. In case fish displayed signs of illness or were needed for routine health monitoring, they were euthanized according to our approved protocol of cold-water shock. PCR-based health monitoring performed by QM Diagnostic (The Netherlands) detected *Mycobacteria spp.* but no presence of *Mycobacterium haemophilum* or *Mycobacterium marinum* in environment swabs or fish samples.

## Structural enrichment

Pilot experiments were performed to determine the most promising form and position of structural enrichment. Different plant forms and sizes were acquired that were certified to be made of non-leaching polyethylene. Four of these aquatic plants were selected that mimic species present in India or Southeast Asia [22] and which were not too large or spiky to fit in our tanks as well as not to pose a danger of injury or swimming impediment to the fish (Dohse Aquaristik, *Hygrophila* #51559, *Vallisneria* #41507, *Rotala* #41514, *Bacopa* #51555). The base of the plants was manually removed as the connecting glue to attach the plants could not withstand hot water for long during the cleaning procedures and might contain leaching chemicals. Instead, a random mix of plastic plants weighing in total about 5 g and of 5–10 cm length were attached to the left tank wall with a rubber suction cup. They were positioned in a way that they stand upright in the water column, that water flow through the tank was not inhibited and no food remains would gather around the plants (Fig 1). In general, the tanks remained easily accessible for cleaning or netting fish. Either one plant bundle was attached in a 3.5 L tank or two next to each other in an 8 L tank (Fig 1C-1F). Every 1–2 months, when a tank needed more thorough cleaning and biofilm removal, fish were transferred to a new tank and received a new plant bundle of similar size but slightly different composition to avoid habituation.

## Rearing

All fish lines used were generated in our quarantine system and needed bleaching of the eggs before being transferred to the main system for rearing. In brief, bleaching was performed on embryos at 24 hours post fertilization (hpf) via two-times 5-minute immersion with 60 ppm sodium hypochlorite based on the ZIRC protocol of "Embryo Surface Sanitation" except with E3 medium for washing instead of E2 medium. 1x E3 medium contained 5.14 mM NaCl, 0.18 mM KCl, 0.34 mM $CaCl_2$ and 0.41 mM $MgCl_2$ and was prepared in 60x stock solution with a pH titrated to 7.2. Embryos and larvae were kept in an incubator set to 28.5°C with 14/10 light-dark hours in petri dishes with up to 50 eggs per 10 cm dish in 20 mL E3 medium. Larvae were checked under a stereo microscope (Olympus SZX16/SZX2-ILLT) to be inconspicuous in morphology and

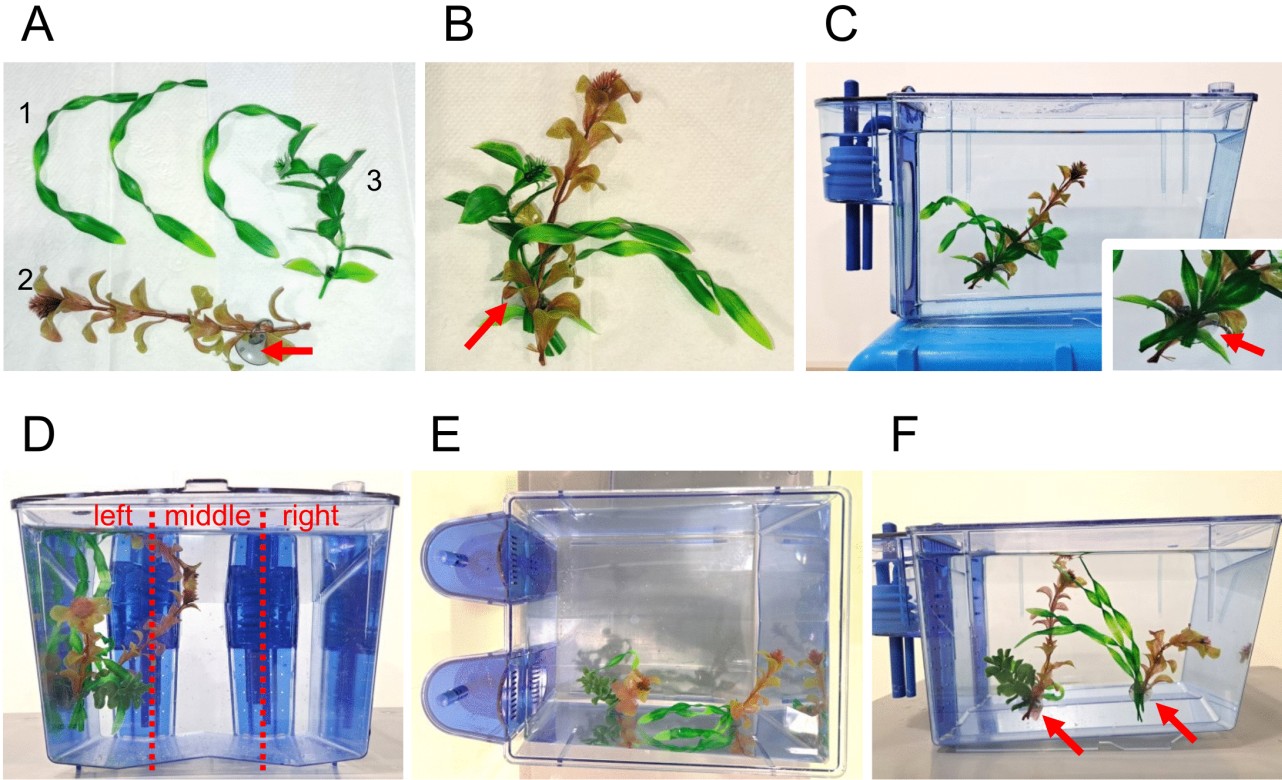

**Fig 1. Artificial plants as structural enrichment. (A)** A combination of artificial plant parts recreating in this example *Vallisneria* (1), *Rotala* (2) and *Hygrophila* (3) were bundled **(B)** and attached to the left tank wall using a half-transparent rubber suction cup (red arrow). **(C)** The position of the plants in a 3.5 L tank is shown with the attachment of the rubber cup enlarged as separate picture. **(D)** The positions of the plants at the left wall in an 8 L tank are shown in front view with a digital separation into left, middle and right area, **(E)** top view and **(F)** side view.

behavior. They were transferred to the main system at 5 days post fertilization (dpf) and randomly distributed to prepared tanks. Either 25 larvae were raised in a 3.5 L tank or 30–37 larvae in an 8 L tank, with their numbers depending on overall needs for breeding these lines in the facility and mating success. All available larvae were equally distributed to tanks with either non-enriched housing conditions (no structural enrichment) or enriched housing conditions, which included artificial plants. All tanks had gravel pictures underneath. Larvae were initially fed only with dry feed (Sparos 100–200), then with increasing fish size feed of larger particle size was added or replaced the smaller sized feed (Sparos 200–400, 400–600). Dropwise artemia feeding started at 10 dpf with increasing amounts during rearing. All lines were raised in the same husbandry rack over the course of a year. They were evaluated for survival, sex ratio and their first two mating results. Tanks of the same line with or without structural enrichment were placed next to each other on the same level in the husbandry rack. The survival rate and sex ratio were determined when the fish were first used for mating at the age of 3–6 months. Detailed dates and numbers for each tank are given in the supplements (S2 Table).

## Mating

Fish aged 3–6 months were used to study effects of structural enrichment in home-tanks during rearing on the first two matings. For mating four 1 L mating tanks were prepared at 4–5 PM with randomly selected 2 females and 1 male each, separated by sex. The mating tanks were then placed on gravel pictures overnight. On the next day at 8.30–9 AM tank water was exchanged and dividers removed. The tanks were placed skewed on gravel pictures to create areas of shallow

and deep water to simulate the shore and facilitate haptic stimulus. Mating was allowed to continue for one hour. After the mating the fish were returned to their home-tanks. The eggs were collected in a strainer and then distributed to petri dishes for counting.

To investigate effects of structural enrichment added to mating tanks only Tübingen wildtype fish were used, aged 4–9 months (tank pairs 3 and 4). On 25 occasions, three tanks were set up for mating as described above, but separated by laminated cardboard between the tanks to prevent visual interaction between fish in neighboring tanks (Fig 2). From these three tanks one tank was placed on an isochromatic colored beige-sand colored tray, the second tank was placed on a gravel picture and the third tank was placed on a gravel picture and a small, swimming plant part was added for enrichment. After 1 hour of mating the fish were returned to their home-tanks and eggs were counted to determine the clutch size.

## Observation of fish/plant interaction

To determine the effects of the structural enrichment on the behavior of the fish, their place preference in the tank with or without artificial plants was investigated. From an initial pilot experiment an effect size of 10% was estimated and under the premise of a power of 90% and an accepted alpha error of 0.05 it was calculated that a sample size of 34 images was needed. With 4 tanks available for each non-enriched or enriched condition (tank pairs # 1–4, containing 26–31 fish per 8 L tank aged 8–13 months), 9 individual pictures per tank were needed for final evaluation. In general, all fish swam calmly and steadily and showed no obvious signs of stress or disturbances when the pictures were taken.

In addition, two situations were distinguished with either ongoing human activity in the husbandry room or with undisturbed fish. Both situations take up roughly 50% of the 13 hours daylight time in our facility on average with more activity in the morning to early afternoon and the undisturbed phase in the late afternoon and evening. In the first case with human presence representing ongoing activity in the facility, a single picture was taken of each tank at 4 PM on 9 different days from 1.5 m distance. For the second case, to observe behavior of undisturbed fish, on a single day a camera was set up at 4 PM at the same spot as before and images were taken automatically with one frame per minute from 5 PM to 9.30

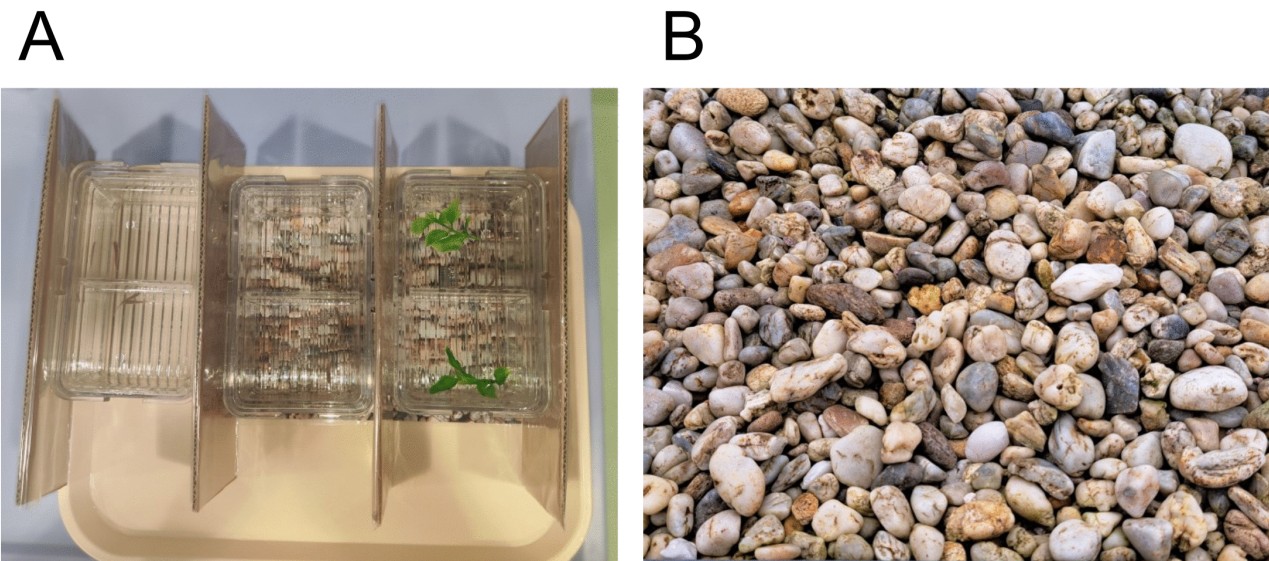

**Fig 2. Structural enrichment during mating. (A)** The effects on mating success and numbers of eggs laid were evaluated using three different conditions for mating tanks that were separated by laminated cardboards: tanks without enrichment, gravel picture only or gravel picture and small plant (from left to right). **(B)** Shown is the gravel picture used for structural enrichment during mating, which was printed in DINA4 and laminated.

PM and starting one hour after the last person left the facility. Then for each tank 9 still images were extracted that were each 30 minutes apart.

In total 144 picture were analyzed with 36 each for non-enriched or enriched tanks with or without human presence. In all cases the positions of fish were first identified as 2D projection of the front view pictures and highlighted as region of interest. They were digitally overlaid with a template to separate the tank equally either in left and right or in left, middle and right areas with plants being at the left wall in enriched tanks (Fig 1D). Then the number of fish per area was counted. In case a fish was located on the border between two compartments, the compartment was chosen where the larger part of the fish body was located. In case equal parts of the fish were in both sections, the one where the head was located was chosen.

Fish from tank pairs 5–8 had been raised in 3.5 L tanks and differed greatly in survival rates determined around 90 dpf as described before. After the first two matings fish from paired tanks were combined to a single enriched 8 L tank at the age of 3–4 months post fertilization (mpf). This resulted in 4 tanks with 28, 32, 44 and 20 fish per tank or holding densities of 3.5, 4.1, 5.5 and 2.5 fish per liter for the tanks 5, 6, 7 and 8, respectively. These tanks were then used when fish were aged 5–7 mpf to investigate effects of holding density on place preference behavior in regards of enrichment as described before. An overview of the sequence of events has been provided in the supplement (S1 Fig).

## Total microbial count

Total microbial counts are performed in our facility on a regular basis to investigate the hygiene status via water samples from the sump. Three times additional samples were taken from the back left corner of two 8 L holding tanks each for non-enriched or enriched conditions of tank pairs 1 and 2 over the course of three months to check for an increased risk of pathogens due to structural enrichment. The samples were diluted by 1:10 and 1:100 with sterile $ddH_2O$ to achieve a countable range of <300 colonies. Sterile water was used as control. 0.1 mL of each diluted sample and control was plated onto a 20 mL sterile PCA (plate count agar) plates (20.5 g/L DM 195, Mast Group #121950) and rested upright for 15 minutes at room temperature until the liquid was absorbed by the plate. The plates were then incubated inverted for 72 +/- 3 hours at 28°C, colonies were counted and number of colony forming units (CFU) per mL were calculated. Experiments with values for water control above 25 CFU/mL were rejected and repeated. Values below 10.000 CFU/mL were considered as safe and below 100.000 CFU/mL as acceptable in regards of fish health and welfare [23].

## Statistics and data analysis

Power calculations were performed based on pilot data to estimate sample size number needed with the NC3R online tool (https://eda.nc3rs.org.uk/experimental-design-group) that used R 3.5.2 and the package power.t.test. In all cases an accepted alpha error of 0.05 and power of 0.90 was selected. Blinding was not possible due to the obvious presence of structural enrichment.

GraphPad Prism 10.1.2 software was used for statistical analysis. Gaussian distribution was tested via Shapiro-Wilk normality test and visually confirmed in a QQ plot. No outliers were identified in our data sets, tested by ROUT method with a maximum desired false discovery rate of 1%. Statistical significance of survival rates, sex ratio and mating success were calculated in a paired t-test. Significant differences between egg numbers were either calculated in an unpaired t-test for the same mating conditions after enriched upbringing or with a One-way ANOVA to test for different enrichments during mating. Differences in place preference between left and right tank side were determined in an unpaired t-test. Statistical significance between place preference of left, middle or right area was calculated via Two-way ANOVA followed by multiple unpaired t-tests or Tukey's multiple comparisons test.

## Ethical approval

As part of our 3R strategy for reducing surplus animals only fish lines already required for other projects or line maintenance were used to study effects of structural enrichment. In addition, only readouts that did not put additional burden

beyond necessary husbandry practices on the fish were selected. Refinement experiments were exempt from approval by local German authorities (LAGeSo) but the zebrafish husbandry, care sheet and euthanasia method of cold-water shock were authorized (ZH 181). Zebrafish were kept in accord with EU directive 2010/63/EU on the protection of animals used for scientific purposes.

## Results

Over the course of one year zebrafish in 8 pairs of tanks have been raised in the same husbandry system, with every tank pair consisting of one non-enriched and an enriched tank with artificial plants. A total of 470 larvae were raised resulting in 360 adults. A detailed results table with fish numbers and their genetic backgrounds is given in the supplements (S2 Table).

Historically the survival rate for zebrafish in our facility determined after rearing at 90 dpf has been 73% on average in the two years before the introduction of structural enrichment started. In this study this value was closely matched with average survival rates of 74% for fish reared again in non-enriched tanks (Fig 3A, 3B). Tanks enriched with artificial plants had no statistically significant improved survival rates of on average 77%. Based on the standard deviation now known to be 11%, the effect size would have had to be 20% to calculate statistical significance with 90% certainty.

Holding density is a known factor to influence sex determination of the zebrafish during rearing [24], and structural enrichment might reduce the available swimming space and thereby the perceived holding density for the fish. The sex ratio was determined around 90 dpf or later during the first two set-ups for matings. In case of tank pair 8 (Tüpfel-Longfin wildtype), only females developed under enriched conditions. This was probably due to their low survival rate resulting in a low holding density which is known to favor female sex determination [25]. Overall, no significant difference was found with an average of 60% and 66% females per tank raised under non-enriched or enriched conditions respectively (Fig 3C, 3D).

One reported benefit of structural enrichment is the reduction of both stress and anxiety for the fish. We hypothesized that this could result in increased numbers of successful matings and egg production, especially on the first two occasions when the fish are not used to the handling and new environment of mating tanks which could potentially increase their stress response. Therefore, fish from paired home tanks that were raised under non-enriched or enriched conditions were set up in parallel when needed for mating. On average the zebrafish raised under enriched conditions showed no better performance in their first two matings with 61 +/- 24% and 82 +/- 19% of tanks having successful matings compared to the 64 +/- 20% and 75 +/- 20% of tanks with fish raised under non-enriched conditions (Fig 4A, 4C). Similarly, no significant difference was seen in the number of eggs laid on these occasions, with on average 68 +/- 55 and 122 +/- 58 laid from enriched fish compared to 72 +/- 43 and 127 +/- 56 eggs laid from not enriched zebrafish (Fig 4B, 4D).

Next, we evaluated if zebrafish in enriched tanks actually showed a difference in their place preference behavior compared to the non-enriched condition. For this, two scenarios were distinguished: About half of the daytime there is ongoing human activity in our facility, with standard husbandry procedures, fish matings and preparation for experiments. In general, the fish don't show any forms of stress in regard to ongoing facility work but tend to stay closer to the front of tanks out of curiosity or in expectation of feeding. When a total of 72 pictures of tank pairs 1–4 were analyzed for place preference of the fish, they evenly occupied both the left and right side in non-enriched tanks (49:51) but avoided the left side with an artificial plant present (36:64) (Fig 5A-5C). On closer inspection, zebrafish in a non-enriched tank preferred the open water of the middle area over wall sides (24:48:28) but rather moved closer to the right wall than to the plant in enriched tanks (14:45:41) (Fig 5D). Similar results were seen for undisturbed fish, less pronounced but still significant with a left:right ratio of 44:56 and 20:47:33 for left:middle:right in enriched tanks (Fig 5E, 5F).

Not all fish known to be in the tank could be counted in a single picture but the number of fish detected was not influenced by the presence of artificial plants, with 83 +/- 9% for both non-enriched and enriched tanks when directly observed (Fig 5C, 5D) and 74 +/-12% and 75 +/- 9% for undisturbed and video-recorded fish under non-enriched or enriched conditions respectively (Fig 5E, 5F).

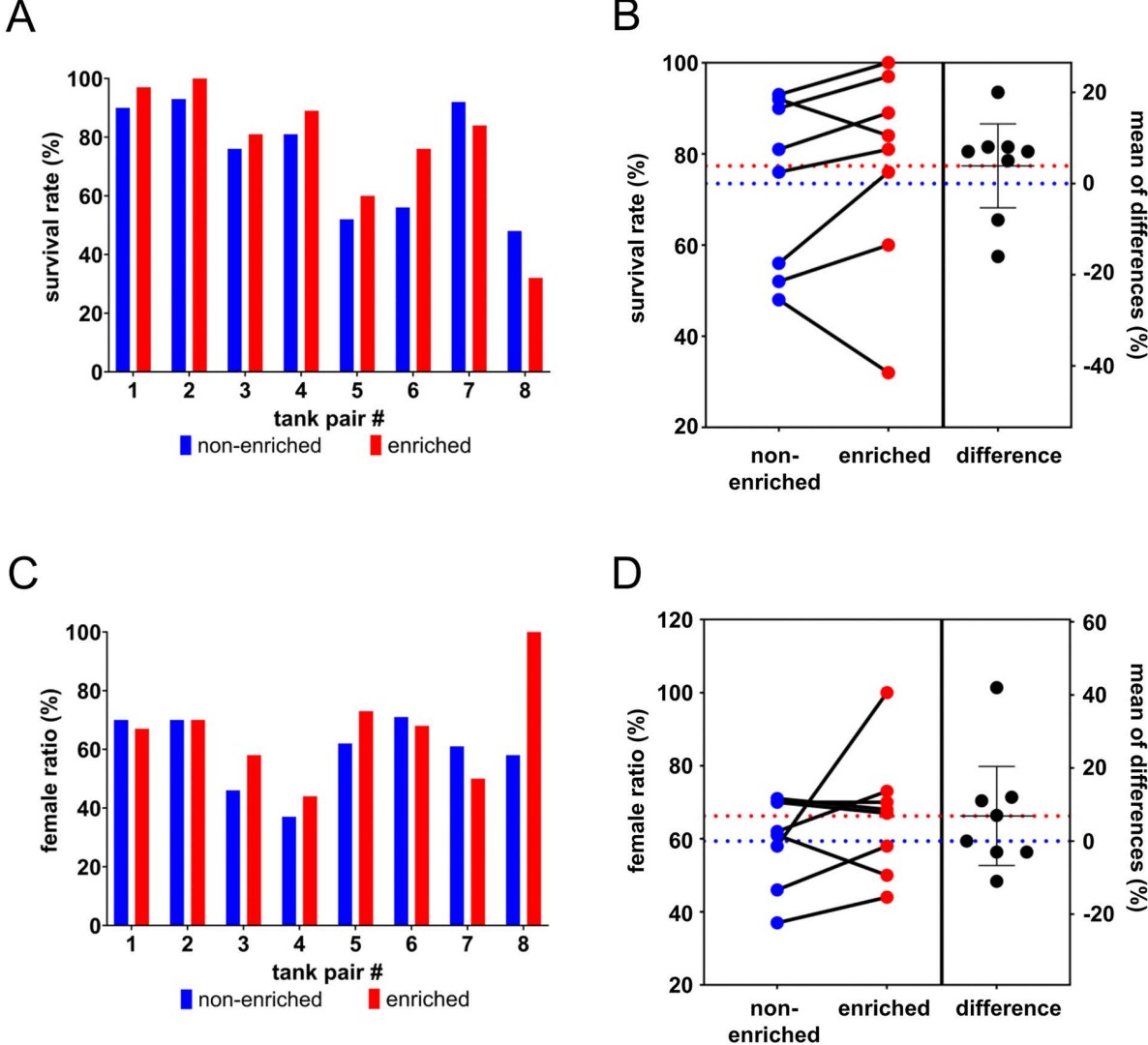

**Fig 3. Influence of structural enrichment on survival rate and sex ratio during rearing. (A)** Eight pairs of tanks with or without structural enrichment were evaluated around 90 dpf for survival rates after rearing. **(B)** The average survival rate for non-enriched tanks was 74% (blue dotted line) and for enriched tanks 77% (red dotted line). A pairwise comparison of the paired tanks (enriched minus non-enriched) showed differences in survival of −16% to +20%, with an average difference of 4% (dotted lines), albeit not statistically significant. **(C)** The ratio of females per tank were determined at the first two matings. **(D)** The average sex ratio was 60% and 66% of females for non-enriched or enriched rearing respectively (dotted lines). The pairwise comparison of enriched tanks and their non-enriched counterparts showed no significant change.

Different survival rates of fish from tanks 5, 6, 7 and 8 resulted in different holding densities of 3.5, 4.1, 5.5 and 2.5 fish/L, respectively. Analysis of place preference in enriched tanks under direct observation showed that in tanks with fewer fish and lower holding density the relative avoidance of the left side with the artificial plant was more pronounced than for fish at higher holding densities (Fig 6A). Only 14% and 7% of fish at densities of 2.5 or 3.5 fish/L stayed in the plant area while at densities of 4.1 or 5.5 fish/L it was 28% and 23%, respectively. In contrast to what was seen before, a majority of over 50% of fish at low holding densities rather stayed close to the right wall, while at higher densities it was only 30–35%. This was also seen with the same fish when they were undisturbed and videotaped, although again to a

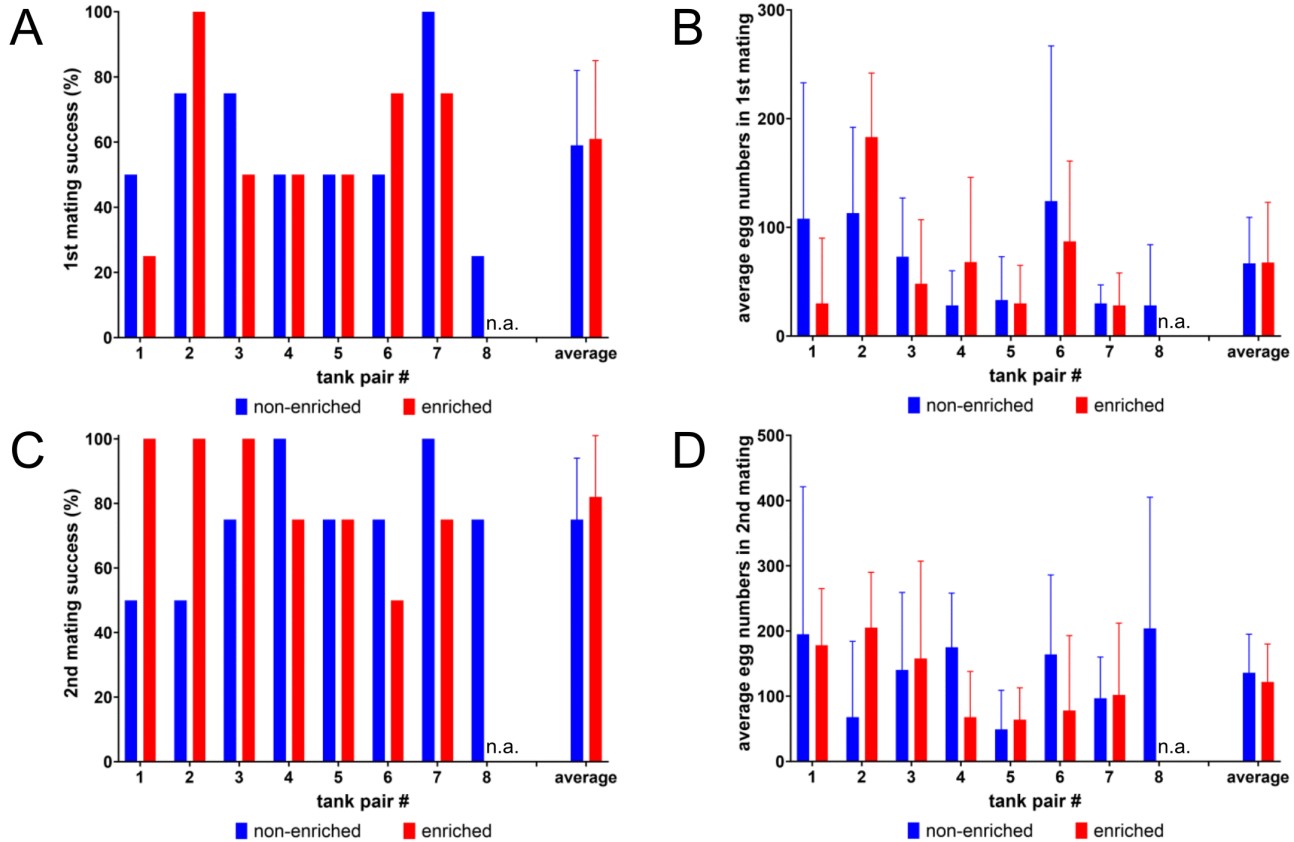

**Fig 4. Influence of structural enrichment on mating success and egg production.** The first two matings that were performed with the fish when they reached adulthood were evaluated based on differences between paired non-enriched (blue) or enriched (red) tanks during rearing. As only female fish were identified in enriched tank 8, no results were available for mating success and egg numbers (n.a.). Neither in the first mating **(A, B)** nor in the second mating **(C, D)** significant differences were seen regarding mating success **(A, C)** or average number of eggs produced per mating **(B, D)**. For the average values the standard deviation is indicated with error bars, n = 7-8.

lesser extent (Fig 6B). In this case, only the fish with the lowest stocking density of 2.5 fish/L showed a different behavior than the others, as with 43% a significantly larger part of them stayed near the right wall, compared to 31–34% from the other tanks.

Another concern with structural enrichment is the increased risk for pathogens. After two months of continuous use, we did not observe biofilm growth on the artificial plants but instead on the comparatively small area of the suction cup. To assess the risk of an overall increased bacterial burden, we analyzed water samples from holding tank pairs 1 and 2 taken at three different weeks via a total microbial count (Fig 7A). The mean number of colony forming units (CFU) per mL were 4633 and 5673 for tank pair 1 and 2387 and 14620 for tank pair 2 with or without enrichment, respectively (Fig 7B). In a repeated measures One-way ANOVA no significant difference was found (p = 0.446). Overall, the values were in the same range as our whole holding system, with a mean of 8843 CFU/mL in the last three measurements of samples from the sump.

Lastly, we wanted to investigate the effect of structural enrichment during mating as another regularly occurring husbandry procedure. Here, only fish of tank pairs 3 and 4 (Tübingen wildtype) were used. In contrast to the former experiment (Fig 4), now all fish were taken from the same home-tank irrespective whether there was enrichment or not and randomly distributed to three different tanks with isochromatic underground (non-enriched), gravel picture or gravel picture

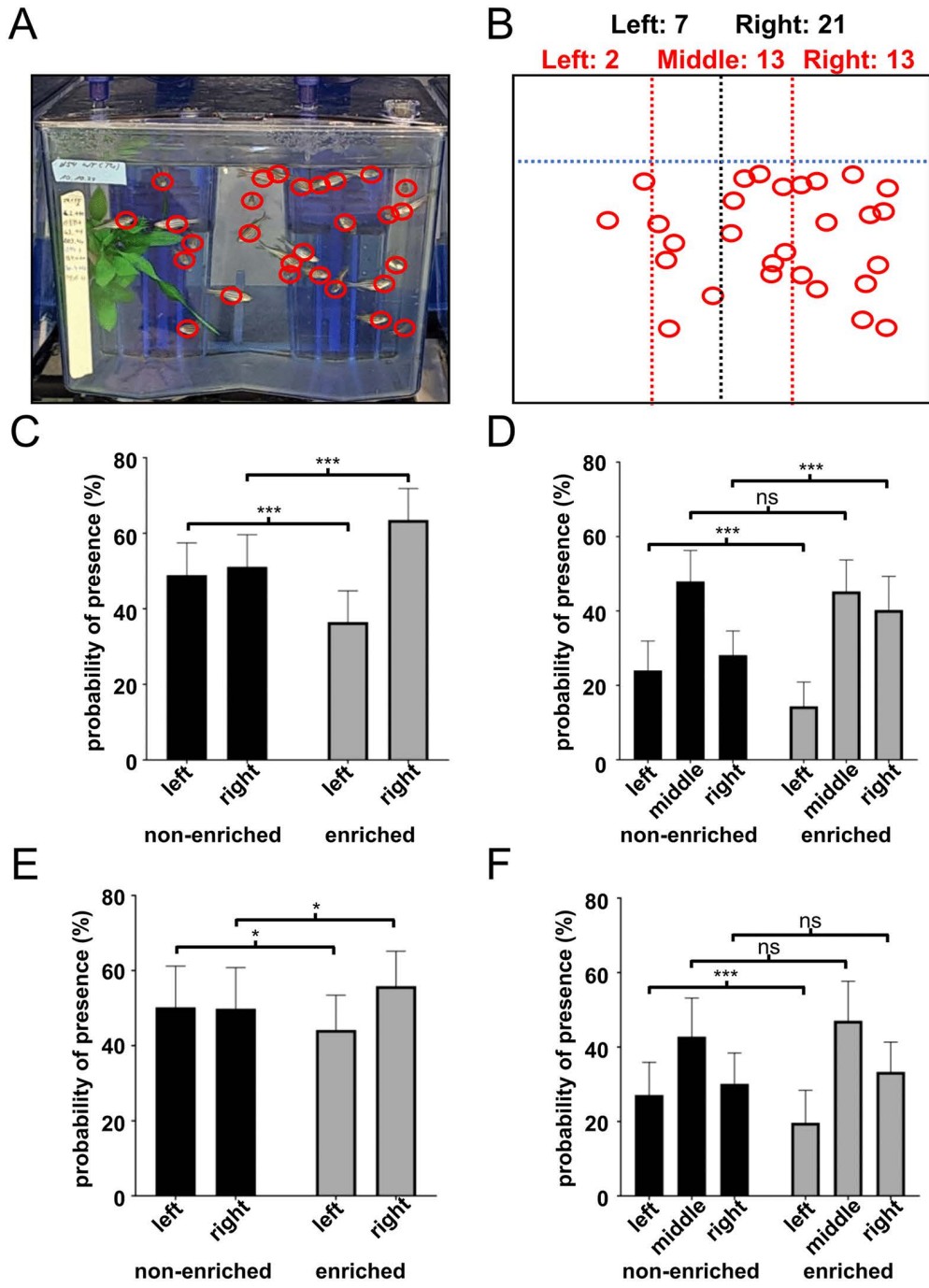

**Fig 5. Effects of structural enrichment on place preference.** Tank pairs 1-4 were investigated for their place preference. **(A)** A representative image is shown of one out of four enriched home-tanks, taken to identify the positions of fish. **(B)** Fish positions (red circles) were overlaid with a digital template to equally divide the tank either in left and right (black dotted line) or in left, middle and right (red dotted lines) areas. The dotted blue line indicates the water level. Then fish per area were counted. **(C)** With human presence fish in non-enriched tanks were divided 49:51 between left and right side, while when enriched, they preferred the free area to the plant area 36:64. **(D)** Divided in three areas the fish in non-enriched tanks preferred the open water in the middle over the left or right area with walls (24:48:28). In enriched tanks they avoided the left area with the plant and stayed more closely to the right wall (14:45:41). **(E)** When undisturbed by human presence, fish were divided 50:50 between left and right side in non-enriched tanks, while with a plant present, they again preferred the free area over the plant area (44:56). **(F)** Of the three areas the undisturbed fish in non-enriched tanks preferred the open water in the middle over the left or right area with walls (27:43:30). In enriched tanks they avoided the left area with the plant, too (20:47:33). Error bars indicate standard deviation, n=36, ns=not significant, * p<0.05, *** p<0.001.

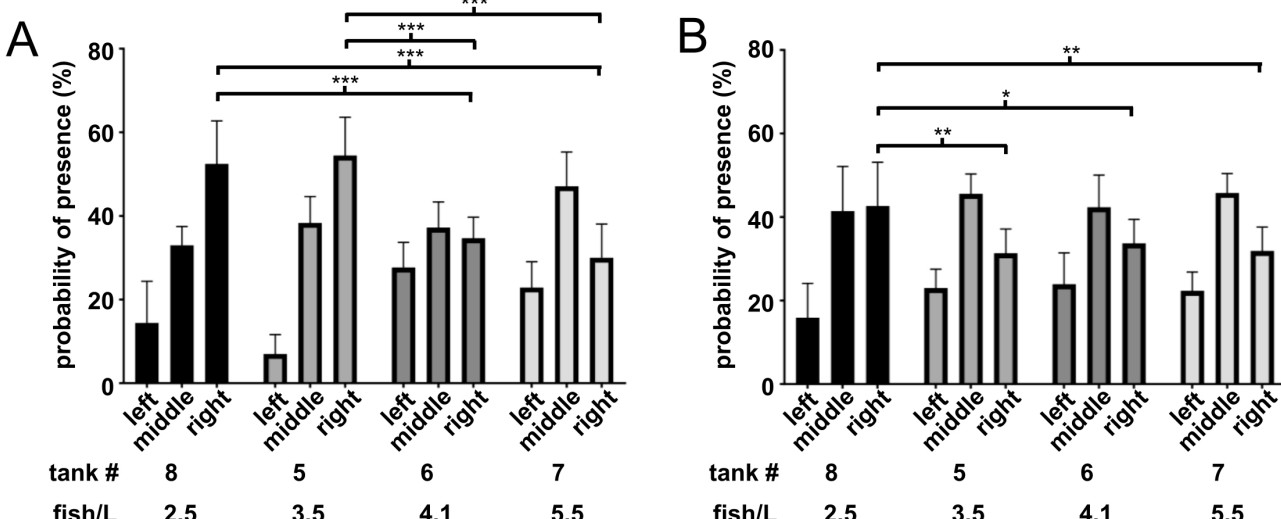

**Fig 6. Effects of holding density on place preference.** Enriched tanks 5, 6, 7 and 8 with different holding densities (fish/L) were observed on 9 occasions for place preference. **(A)** With human presence fish with lower holding densities of 2.5 or 3.5 fish/L had an increased avoidance of the enriched left tank side and a stronger preference for the right wall side. **(B)** Undisturbed fish with the lowest holding density of 2.5 fish/L still preferred the right wall side compared to tanks at higher holding densities. Error bars indicate standard deviation, n = 9, * $p < 0.05$, ** $p < 0.01$, *** $p < 0.001$.

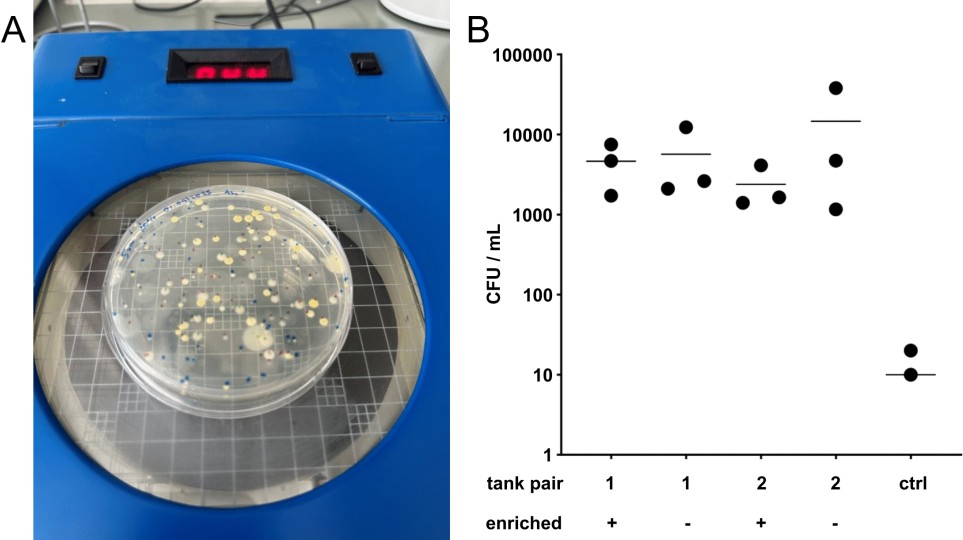

**Fig 7. Effects of structural enrichment on the risk of pathogens. (A)** Water samples were taken from the left bottom corner at the back of the tank pairs 1 and 2 on three different occasions and plated on PCA to determine the total microbial count. Sterile water was used as control. **(B)** No significant difference in CFU/mL was observed in tanks with (+) or without (-) enrichment. The third count for the water control (ctrl) was 0 which cannot be displayed on a logarithmic scale. The horizontal bars indicate mean values, n = 3.

plus plant (Fig 2). A printable version of the gravel picture has been provided as supplement (S2 Fig). On average 149, 124 or 135 eggs were laid per mating for non-enriched conditions with an isochromatic beige-sand underground, enrichment by gravel picture under the mating tank or an artificial plant inside the tank in addition to gravel pictures, respectively

(Fig 8). No significant differences were found based on calculation of repeated measures One-way ANOVA between any of these conditions (p = 0.564).

## Discussion

Aside from ethical and legal obligations there is consensus in the scientific community that laboratory animals need constant improvement of husbandry conditions to allow for optimal data quality [26]. In addition, lack of standardization is also a concern for the reproducibility of research data generated with fish under different environmental conditions [26]. It has been argued that structural enrichment actually increases data robustness because fish are more resilient even when less standardized objects are used [1]. In the case of zebrafish, there is an ongoing discussion whether artificial plants in husbandry systems could improve animal welfare. Currently only a minority of husbandry facilities use structural enrichment or they are at least not reporting it according to the latest survey of Lidster et al. and the 100 most recent zebrafish publications we reviewed [9]. Named reasons were concerns regarding unclear benefits, costs, risk of pathogens, chemical leaching, or increased aggression [9,15]. For structural enrichment to be widely accepted and implemented in husbandry facilities, the benefits need to clearly outweigh the drawbacks. Nevertheless, the improvement of fish welfare can be challenging to measure directly. In addition, long-term effects are more relevant for husbandry but currently only few studies addressed these [12]. Still, the first systematic review on effects of structural enrichment for zebrafish found that despite many confounding factors, overall, the positive effects are holding up [12].

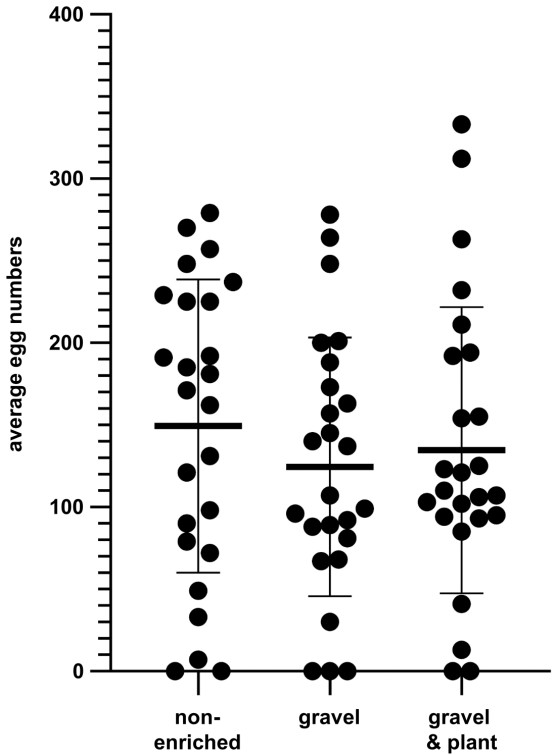

**Fig 8. Influence of structural enrichment during mating.** Wildtype Tübingen fish were repeatedly set up for mating with different enrichments and the resulting number of eggs were counted. On average neither the gravel picture under the tanks alone nor together with artificial plants in the mating tanks resulted in a significant change of egg numbers produced during matings. Dots represent individual results; horizontal bars mean values. Error bars indicate standard deviation, n = 25 matings per group.

Therefore, we decided to introduce artificial plants as a new form of structural enrichment in our zebrafish facility and to monitor its effects in comparison to our standard husbandry setup for one year. We have chosen plants made of certified non-leaching polyethylene, with low material costs of 3–5 Euro per tank, adding up to less than 100 Euro for the whole rack, depending on the mix of plants used. After one year of more or less continuous use and including several dish washer cleanings at 80 °C, the material showed no sign of deterioration and we expected the plants used to last for several years. We were able to place the plants in the tanks in a way that they didn't obstruct water flow or daily inspections of the fish, as demonstrated by similar detection rates for fish in pictures and movies taken from the tanks during place preference analysis. We found no increased biofilm growth on the plants and could not detect an increase of the total microbial count in the water of the enriched tanks, indicating that UV light disinfection of our holding system was sufficient to keep microbial burden low [23,27]. In the observed time period, no fish needed to be removed from tanks due to signs of illness. Therefore, it remains unclear if enrichment improved health or hindered identification of sic fish. Overall, we concluded that the chosen form of enrichment was low cost, easy to introduce and not acting detrimentally on the water quality of our system or to the health of the fish.

We have not systematically investigated the effects of territorial behavior or aggression on the fish. We have occasionally observed that single fish remained near the plants and chased other incoming fish away, indicating a high place preference for these individual fish. This behavior never lasted until the next day and in the meantime did not seem to affect the behavior of the remaining shoal that kept calmly swimming in the open water area. There have been conflicting reports on how structural enrichment affects aggressive behavior and it needs to be further investigated in the future what combination of factors like holding density or the exact form of enrichment are inducing or preventing aggression [2,15,28–30].

From our experience, the main drawback for implementing structural enrichment was a small but unavoidable additional work load for our animal caretakers which we estimated to be one person-hour per month for a complete rack with 50 enriched tanks. This includes preparing or replacing appropriate structural enrichment, its maintenance in the tank, like re-attaching loosened structures or rearranging structures inside the tank during cleaning, and cleaning the plants after use.

To determine benefits of structural enrichment, we used mainly readouts that are most important for our work as a facility and were therefore already established and continuously evaluated anyway. When comparing our new lines raised in regards of survival, sex ratio, mating success and numbers of eggs laid with or without enrichment no statistically significant effect was found. The survival rate was especially low in the last case observed (enriched tank 8, Tüpfel-longfin wildtype). This was likely due to a bad quality of this clutch as the parent line was already old and had several unsuccessful matings before a small number of eggs was obtained. These eggs were below standard quality and had to be raised nonetheless to maintain the line. This case might not be comparable with the other lines but we decided against excluding its data as it represents usual problems in a husbandry facility. Unfortunately, this resulted in a standard deviation much higher than initially anticipated and therefore we were unable to gather enough samples to reach statistical significance. This is in contrast to a considerable improvement of 29% in the survival rate reported by Lee et al. 2019, which was likely only possible as only a single clutch was investigated. It has to be considered that in their study the survival rate of their non-enriched conditions was only 54% and with enrichment it reached 83%, which is a similar level we have seen in our lab with non-enriched conditions [8]. In general, as we had already established very high hygiene standards, excellent water quality that is constantly monitored, and enrichment in the form of gravel pictures under all tanks, it might be expected that the addition of artificial plants would not show such substantial improvements as seen in the study by Lee at al. 2019. Still, we would consider the average improvement of 4% in survival rate of our fish from enriched tanks as biologically relevant, if this were confirmed in a larger data set.

Another study showed that zebrafish shoals raised with structural enrichment for 6 months were less anxious [31]. In regards to fertility, we hypothesized that addition of artificial plants could help to reduce anxiety and stress and thereby improve mating success and egg laying, but we saw no effect either with plants added to the home-tanks or to the mating

tanks. This is mostly in agreement with Woodward et al. who saw no differences in most of their settings, too [15]. In contrast, Wafer at al reported a strong increase of egg production due to structural enrichment [32]. One reason for this could be a difference in the initial stress levels. If the stress level was already low under non-enriched conditions, it might not be significantly reduced any further. When Sen Sarma et al. investigated the effect of stocking densities on stress levels, they saw high cortisol levels at a very low density of 1 fish/L. The cortisol levels were strongly reduced at 3 or 6 fish/L which reflects better the situation in our husbandry [2]. In our own experience, young and healthy fish (3–12 months post fertilization, mpf) mate easily and very successful even in barren environments. But for older fish or lines with higher degrees of inbreeding that would otherwise be less fertile, enrichment in the form of gravel pictures or providing areas with shallow and deeper water in the mating tanks can improve chances for a successful mating. Therefore, it would be interesting to investigate in the future whether addition of artificial plants could improve health and fertility in older fish (> 15 mpf).

So far, the strongest evidence in favor of structural enrichment comes from place preference experiments that showed that zebrafish would rather occupy higher structured areas than barren tanks [6]. At the same time this highly social fish species displays strong shoaling behavior which can only be performed in open water [33]. By observing wild caught zebrafish in a large arena, it has been suggested that the preference for more vegetation could even outweigh the desire for larger groups [34]. In addition, overall preference for enriched environment was not changed by domestication of laboratory zebrafish compared to wild-caught zebrafish, but detailed comparisons are rare and merit further investigations [35]. Given the current commercial tank sizes of up to 10 liters, it remains a challenge to provide both large shoals and a rich environment as found in the zebrafish's natural habitat with its extensive ponds and rivers.

Therefore, we wanted to examine the long-term effects of structural enrichment in regards of place preference in our holding tanks. In addition, an animal facility can be a busy place and we examined how human presence might affect this preference. We found that under non-enriched conditions the fish had no preference for the left or right tank side but instead preferred the open water in the middle over the areas with side walls. When artificial leaves were placed at the left side of the tank, fish were significantly less likely to be present in that area but instead moved more to the middle compartment or the previously less favored right wall. This effect was seen both with and without human presence but more pronounced in the first case. Here, even a distant perceived human presence was sufficient to attract the fish closer to the front out of curiosity or in expectation of feeding. The extent of this effect depended on holding densities as well, with increasing fish numbers correlating to their presence close to plants. We concluded that only when fish seemed unwilling to further increase their shoal cohesion and needed to maintain an average distance to all other fish in their vicinity as well, they entered enriched areas. When undisturbed, the distribution of fish with enrichment was more similar to non-enriched conditions. A possible explanation could be that undisturbed fish swim slower on average and are therefore able to move closer to the artificial plants or were occupying more often rear parts of the tanks, resulting in a less pronounced change of place preference. The remaining difference to non-enriched conditions might be just the physical volume the plants occupy and not an induced change in fish behavior. This needs to be investigated in more detail in the future. Still, in contrast to the initial working hypothesis, the artificial plants were clearly not able to attract more fish to their closer vicinity but in the best case fish were indifferent to their presence and in the worst case fish were displaced to areas usually avoided due to restriction of free swimming space.

One limitation in our approach was, that the tanks were not visually separated in order to avoid major changes in our husbandry setting. Therefore, plants in enriched tanks were only partly obscured and fish from the non-enriched tanks were likely able to see them. Interaction is not always necessary to fulfill animal needs [36]. In the case of zebrafish, it was shown that gravel pictures are nearly as attractive as gravel itself [13] and images of siblings alone could attract zebrafish [37]. It is thereby possible that plants acted not as structural but as visual enrichment which might have limited the effect size in paired tanks for mating success or eggs numbers in the first two matings. It merits further investigation, if additional pictures outside the tanks can improve zebrafish welfare as this would circumvent many concerns for placing material inside the tanks.

Overall, further research is needed what structures and forms of enrichment improve animal welfare for fish at holding densities and numbers that are typical for husbandries. It should also reduce the likelihood of hierarchies developing among the fish or at least provide sufficient hiding places from dominant or aggressive fish. In addition, the measurable health benefits, such as an improved survival rate for larvae or fertility in older fish, must outweigh the risk for overlooking sick fish during visual inspections. Based on our findings of shoaling place preference, fish seemed to prefer open water over areas close to tank walls. This raises the question if larger tanks are always preferable or what tank volumes and dimensions provide optimal animal welfare depending on the contained number of fish. Lastly, the potential for cognitive improvement even when fish only occasionally interact with structural enrichment, should be investigated in more detail, as there is already evidence that raising laboratory or wild-caught zebrafish under enriched conditions improves their learning abilities [10,35,36,38].

In summary, we think that in previous publications lack of standardization and major differences in husbandry procedures and systems resulted in different outcomes or a variable extent of the effects of structural enrichment. We found that introducing structural enrichment in our facility was low cost and had no detrimental effects on the fish but showed no significant benefits either. Here, additional data with a larger sample size would be needed to determine these more subtle effects which our small facility could not provide yet. Therefore, we would encourage other facilities to test new applicable forms of enrichment, too and report their findings for their specific settings to further an open discussion regarding good practices. Most importantly fish behavior in the form of aggression or place preference needs to be studied in long-term observations in more detail in the future. As previous research on older animals suggested, a more structured environment could be especially important to maintain health and cognitive functions at advanced age.

## Supporting information

**S1 Table. PubMed Search list.**
(XLSX)

**S2 Table. Raw data tables.**
(XLSX)

**S1 Fig. Timeline of events.**
(DOCX)

**S2 Fig. Gravel picture for printing.**
(JPG)

## Acknowledgments

We thank our animal care takers for maintenance of BfR zebrafish facility, providing excellent husbandry conditions and their support in collecting the data. We thank the whole team of the Aqua facility and Dr. Stefanie Banneke for helpful suggestions and discussions.

## Author contributions

**Conceptualization:** Nils Ohnesorge.

**Data curation:** Nils Ohnesorge.

**Formal analysis:** Aymene Youcef Krachni, Jenny Wilzopolski, Nils Ohnesorge.

**Investigation:** Aymene Youcef Krachni, Richard Busch, Indigo Brakus, Angelina Schumann, Nils Ohnesorge.

**Methodology:** Jenny Wilzopolski, Nils Ohnesorge.

**Project administration:** Nils Ohnesorge.

**Supervision:** Jenny Wilzopolski, Nils Ohnesorge.

**Visualization:** Nils Ohnesorge.

**Writing – original draft:** Nils Ohnesorge.

**Writing – review & editing:** Jenny Wilzopolski, Nils Ohnesorge.

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
