## [Decision Letter · Decision Letter 0]

23 Jan 2026

PONE-D-25-66088Zebrafish facility report on implementation of artificial plants as structural enrichmentPLOS One

Dear Dr. Ohnesorge,

Thank you for submitting your manuscript to PLOS ONE. After careful consideration, we feel that it has merit but does not fully meet PLOS ONE’s publication criteria as it currently stands. Therefore, we invite you to submit a revised version of the manuscript that addresses the points raised during the review process.

Both reviewers raise similar points, many of them concerning the interpretation and wording of your results. As one reviewer pointed out, the wording of the main headings are particularly important, since they are often read by people involved in regulatory offices with minimal scientific background.

We look forward to receiving your revised manuscript.

Kind regards,

Stephan C.F. Neuhauss, Ph.D.

Academic Editor

PLOS One

2. To comply with PLOS One submissions requirements, in your Methods section, please provide additional information regarding the experiments involving animals and ensure you have included details on (1) methods of sacrifice, (2) methods of anesthesia and/or analgesia, and (3) efforts to alleviate suffering.

3. Please note that PLOS One has specific guidelines on code sharing for submissions in which author-generated code underpins the findings in the manuscript. In these cases, we expect all author-generated code to be made available without restrictions upon publication of the work. Please review our guidelines at >https://journals.plos.org/plosone/s/materials-and-software-sharing#loc-sharing-code and ensure that your code is shared in a way that follows best practice and facilitates reproducibility and reuse.

Reviewers' comments:

Reviewer's Responses to Questions

**Comments to the Author**

1. Is the manuscript technically sound, and do the data support the conclusions?

Reviewer #1: Partly

Reviewer #2: No

2. Has the statistical analysis been performed appropriately and rigorously? 

Reviewer #1: Yes

Reviewer #2: I Don't Know

3. Have the authors made all data underlying the findings in their manuscript fully available?

Reviewer #1: Yes

Reviewer #2: Yes

4. Is the manuscript presented in an intelligible fashion and written in standard English?

Reviewer #1: Yes

Reviewer #2: Yes

5. Review Comments to the Author

Reviewer #1: This is a fascinating manuscript looking at the effect of use of artificial plants in zebrafish facitlities. It has some very useful results but needs significant modification as the conclusions of the paper poorly reflect the data.

I have the following comments

Line 20/21 abstract: “It was shown that this can reduce stress and improve cognitive abilities”..this was a bit confusing as I expected that the paper itself generated this data, which was not the case…please change to “Other reports have…

Line 27 avoid stating increases if not statistically significant, it should be: No statistically increase of survival was observed, also mention absence of significant benefit in terms of egg laying. Mention that the avoidance of plants was statistically significant.

I would weaken the final statement in the abstract in the light of the data presented by the paper itself.

Line 154: Plants that are present in the natural habitat of zebrafish: reference needed, this seems very difficult to verify, amend?

Figure 1: Position of plant in 8 liter tanks should ideally also be shown in this figure (a top down view would be best), the rubber suction cup is not visible in C, but the main text suggests otherwise, please show detail, or a separate picture.

Line 181 With less larvae available? Odd do you mean: If less than 25 larvae per tank were available…?

Line 186 Were the fish in the non-enriched tanks able to see the plants in the enriched tank? Could this visual stimulation have effects? It might have been better to visually isolate them. This might need some discussion.

Line 218 the picture regime in the quiet time is clear and makes sense however the regime during the day is unclear it sounds like only one picture was taken at 4pm, which will unlikely be sufficiently representativie of activity during the day..please make this more explicit

Line 223 “separate the tank equally either in left and right or in left, middle and right areas.” I understand from this that a 2D projection and not a 3D position was determined, this need to be made explicit, or clarified better, it also need to be clear where plants are in this setup. I realise there is a picture in Fig5..refer to this at this point in the text?

Line 237 The precise raising and pooling strategies are hard to follow from the text, please add a arrow diagram/timeline in the supplementary data.

Line 284 Phrasing is misleading, here there is no significance. This needs to be changed to: The addition of plastic plants did not lead to a statistically significant increase or decrease in survival. Then provide calculations of which difference would be detectible with 90% security, using your spread in the data as a guide. (NB also the fact that enriched had better survival 6 out of 8 times is not statistically significant (>14%). In short, so give some indication of the sensitivity of your essay

Line 288 The sex of the zebrafish is determined mostly by environmental factors: I do not think this is true, there are sex influencing loci as well, amend and/or provide references for such statements. (N.B. Here phrasing is neutral, as it should be.)

Line 297 As mentioned before (line 284) use neutral phrasing as is done for panel D.

Discussion and overall conclusions:

In the final discussion it is made clear that there are no significant effects from the plants, apart from the fact that the fish are shown to avoid the plants. As an argument in favour of using them despite this, other papers are used (eg ref 14), I disagree with this way of drawing conclusions as it based on data that are gathered often under very different conditions, often testing fish in isolation, which is not what happens in stock maintenance.

The discussion and abstract advocate measures that are not supported by the author’s own measurements. Even though the required effort may be considered small, introducing plastic plants across thousands of tanks in 100s of facilities should only be supported if advantages are directly observed, with clear statistical significance.

Even more importantly, only thing that does show a significant result, is avoidance of the plants, to me this indicates that the fish do not like them and it causes artificial crowding. Place preference has been used extensively, as a way to measure aversion in behavioural studies, including several studies in mentioned in ref 14. I take from the data that lab-bred zebrafish maybe are adapted and are (mis)perceiving plants as a threat rather than an enrichment. This is surprising and interesting, and it would be extremely interesting to know if newly wild-caught zebrafish show the same behaviour, it would be very interesting to collaborate with labs (eg in india) who would have acces to such fish. This could be part of a discussion

Plants hinder visual health checks; the discussion should weigh the (un)measurable positive enrichment effect against the negative effect of (even if this happens rarely) missing a sick fish in visual inspections that may then perish and experience severe suffering.

Overall, I conclude that this study is certainly worthy of publishing however the tone of the article needs reflect the data that the paper obtains, and I think it should have the following title, which fits the data that were produced by the paper:

“Introduction of artificial plants does not give significant benefit during raising and breeding, but causes an aversive response in laboratory strains of zebrafish housed under standard conditions”

It is very important that the headline of this paper is right, because it may be read by people who make high-impact policy decisions, but do not go beyond title/abstract level.

Reviewer #2: This paper by Krachni and colleagues presents a well-motivated facility study showing that artificial plants provide a non-detrimental form of enrichment, with subtle/no benefits for survival, egg production, and sex ratio. The long-term implementation in a real facility over one year is valuable and relatively rare. Environmental parameters and husbandry conditions are described in detail. However, several aspects of the study need improvement for the work to be of broader interest to the community:

1. The Introduction is quite long and repeats several general arguments about enrichment, results from previous studies and lack of data. Consider revising to improve clarity and impact.

2. This study is based on the usage of different genetic lines introduced in Methods. However, in the results these are named “line 1”, “line 2”, etc. without specifying what they are or without interpreting results in the light of these different genotypes. If these groups are genetically distinct, then we would expect some reference to what they are alternatively the term “cohort” or “group” would be a more appropriate to describe them.

For exapample in the discussion: “The survival rate was especially low in the last case observed (line 6). This was likely due to a bad quality of this clutch as the parent line was already old and had several unsuccessful matings before a small number of eggs was obtained. These eggs were below standard quality and had to be raised nonetheless to maintain the line.”…What is line 6? Does this correspond to a specific genotype? What is the age of these fish? Can line 6 be compared with another line of the same age that showed a better outcome?

3. In addition, the manuscript alternates between “lines,” “tanks,” and “conditions,” which make the experimental design difficult to follow. Using consistent and clearly defined terminology would greatly improve clarity.

4. The authors should indicate in the text the total number of fish per “line/tank” and/or condition. As it stands, the frequent reference to ‘lines 1–6’ without clear reference to actual animal numbers makes it difficult to appreciate sample size and study scale

For example “Eight tanks of lines 1 and 2”, What does this indicate? four each, 2/6 or 8 each? Also” Over the course of one year six different lines have been raised in 16 tanks in the same husbandry system. Of every line half the tanks received structural enrichment in the form of artificial plants while the other half were kept under our standard conditions.” Here we have no idea of tank distribution per line and how many fish…."

5. Statements such as ‘in 6 of 8 cases the survival rate was better with structural enrichment’ read somewhat anecdotal, especially in the absence of statistical significance. It would be preferable to focus on effect sizes, and to avoid more colloquial formulations that may overstate the value of a trend that is statistically non‑significant.

6. The comparison between “standard” and “plant-enriched” environments is potentially confusing, as “standard” is vague and context-dependent, and may imply an optimal condition rather than a descriptive one. A clearer formulation would be: “Fish were raised under either non-enriched housing conditions (no structural enrichment) or enriched housing conditions, which included artificial plants.” This wording would avoid ambiguity.

7. Importantly, this study distinguishes several genetic lines, each split into tanks with and without plants, which is sensible biologically. However, this fragments the data and further reduces power. Given that the main question is whether plant enrichment affects outcomes at all, it would be informative to analyse all tanks together as ‘enriched’ vs ‘non‑enriched’, while including line as a factor (or random effect) rather than running effectively separate mini‑studies per line (without however taking genotypes into account for interpretation). The current presentation, where ‘lines’ and ‘tanks’ are repeatedly described but rarely summarised into a single enrichment effect across lines, makes it harder for the reader to see the overall impact of enrichment on the fish. Re‑analysing the data with all tanks grouped as ‘enriched’ vs ‘non‑enriched’, and treating line as a factor, is unlikely to change the general conclusion (no strong effect), but it would provide a clearer and statistically more robust estimate of the overall enrichment effect than the current fragmented, line‑by‑line presentation. Have the authors consider this? Can they comment on this?

8. Linked to the previous point, I am not an expert here, but to my understanding survival, sex ratio, and mating success are binomial (yer/no), and the use of t-test–based power calculations is not optimal for such bounded data, especially with small numbers of tanks; statistical methods designed for proportions would be more appropriate.

9.Behavioral conclusions about welfare in particular stress are largely indirect because no direct measures were collected. Considere tuning down

6. PLOS authors have the option to publish the peer review history of their article (what does this mean?). If published, this will include your full peer review and any attached files.

Reviewer #1: No

Reviewer #2: No

---

## [Author Response · Author response to Decision Letter 1]

9 Mar 2026

Dear Editor and Reviewers,

Thank you for reviewing our manuscript PONE-D-25-66088 and providing helpful comments and suggestions. We feel that it has now significantly improved and hope for your acceptance. Please see the detailed answers to the points raised in the file attached.

With best regards,

Nils Ohnesorge

---

## [Decision Letter · Decision Letter 1]

20 Apr 2026

Introduction of artificial plants had no detrimental or beneficial effects on laboratory zebrafish husbandry but limited available swimming space

PONE-D-25-66088R1

Dear Dr. Ohnesorge,

We’re pleased to inform you that your manuscript has been judged scientifically suitable for publication and will be formally accepted for publication once it meets all outstanding technical requirements.

Please phrase the title in present tense ("Introduction of artificial plants has no detrimental or beneficial effects on laboratory zebrafish husbandry but limited available swimming space").

Kind regards,

Stephan C.F. Neuhauss, Ph.D.

Academic Editor

PLOS One

Additional Editor Comments (optional):

Reviewers' comments:

Reviewer's Responses to Questions

**Comments to the Author**

1. If the authors have adequately addressed your comments raised in a previous round of review and you feel that this manuscript is now acceptable for publication, you may indicate that here to bypass the “Comments to the Author” section, enter your conflict of interest statement in the “Confidential to Editor” section, and submit your "Accept" recommendation.

Reviewer #1: All comments have been addressed

Reviewer #2: All comments have been addressed

2. Is the manuscript technically sound, and do the data support the conclusions?

Reviewer #1: Yes

Reviewer #2: Yes

3. Has the statistical analysis been performed appropriately and rigorously? 

Reviewer #1: Yes

Reviewer #2: Yes

4. Have the authors made all data underlying the findings in their manuscript fully available?

Reviewer #1: Yes

Reviewer #2: Yes

5. Is the manuscript presented in an intelligible fashion and written in standard English?

Reviewer #1: Yes

Reviewer #2: Yes

6. Review Comments to the Author

Reviewer #1: I am happy that the authors have taken the comments to heart and support publication, it is a valuable addition to the ongoing enrichment debate. The title has one grammatical issue , cthe tense should the "present".

Introduction of artificial plants has no detrimental or beneficial effects on laboratory

zebrafish husbandry but limited available swimming space

Reviewer #2: The authors have sufficiently addressed the reviewers’ comments, and the manuscript has improved in clarity and overall quality. I support the publication of this interesting study.

7. PLOS authors have the option to publish the peer review history of their article (what does this mean?). If published, this will include your full peer review and any attached files.

Reviewer #1: No

Reviewer #2: No

---

## [Editor Report · Acceptance letter]

PONE-D-25-66088R1

PLOS One

Dear Dr. Ohnesorge,

I'm pleased to inform you that your manuscript has been deemed suitable for publication in PLOS One. Congratulations! Your manuscript is now being handed over to our production team.

Kind regards,

on behalf of

Dr. Stephan C.F. Neuhauss

Academic Editor

PLOS One